# Controlling the Carrier Injection Efficiency in 3D Nanocrystalline Silicon Floating Gate Memory by Novel Design of Control Layer

**DOI:** 10.3390/nano13060962

**Published:** 2023-03-07

**Authors:** Hongsheng Hu, Zhongyuan Ma, Xinyue Yu, Tong Chen, Chengfeng Zhou, Wei Li, Kunji Chen, Jun Xu, Ling Xu

**Affiliations:** 1School of Electronic Science and Engineering, Nanjing University, Nanjing 210093, China; 2Collaborative Innovation Center of Advanced Microstructures, Nanjing University, Nanjing 210093, China; 3Jiangsu Provincial Key Laboratory of Photonic and Electronic Materials Sciences and Technology, Nanjing University, Nanjing 210093, China

**Keywords:** C–V memory window, nanocrystalline Si, floating gate memory

## Abstract

Three-dimensional NAND flash memory with high carrier injection efficiency has been of great interest to computing in memory for its stronger capability to deal with big data than that of conventional von Neumann architecture. Here, we first report the carrier injection efficiency of 3D NAND flash memory based on a nanocrystalline silicon floating gate, which can be controlled by a novel design of the control layer. The carrier injection efficiency in nanocrystalline Si can be monitored by the capacitance–voltage (C–V) hysteresis direction of an nc-Si floating-gate MOS structure. When the control layer thickness of the nanocrystalline silicon floating gate is 25 nm, the C–V hysteresis always maintains the counterclockwise direction under different step sizes of scanning bias. In contrast, the direction of the C–V hysteresis can be changed from counterclockwise to clockwise when the thickness of the control barrier is reduced to 22 nm. The clockwise direction of the C–V curve is due to the carrier injection from the top electrode into the defect state of the SiN_x_ control layer. Our discovery illustrates that the thicker SiN_x_ control layer can block the transfer of carriers from the top electrode to the SiN_x_, thereby improving the carrier injection efficiency from the Si substrate to the nc-Si layer. The relationship between the carrier injection and the C–V hysteresis direction is further revealed by using the energy band model, thus explaining the transition mechanism of the C–V hysteresis direction. Our report is conducive to optimizing the performance of 3D NAND flash memory based on an nc-Si floating gate, which will be better used in the field of in-memory computing.

## 1. Introduction

With the development of the Internet in modern society, highly efficient processing of big data is faced with the challenge of breaking up the storage walls and power consumption walls induced by traditional von Neumann architecture. Recently, in-memory computing has attracted great interest, due to its ability to effectively improve the processing efficiency of big data. As a strong device candidate for in-memory computing, silicon-based 3D NAND flash memory with perfect compatibility with CMOS technology has attracted much attention [1,2,3,4,5]. However, the traditional 3D NAND flash, based on a polysilicon floating gate, is confronted with the current leakage problem, which is induced by random defects in the tunnel oxide layer after repeated erasing and writing [6,7,8]. As the charges stored in the floating-gate layer can be free to move laterally, it results in data loss. To solve the current leakage problems, the nanocrystal has been adapted to the floating-gate memory [8,9,10,11]. In contrast to the polysilicon floating gate, using discrete nanocrystals as the charge storage layer can avoid the free movement of charges laterally in the floating-gate layer, thus effectively preventing data loss caused by charge leakage [12,13,14]. In addition, the ultra-thin tunnel oxide layer in the nanocrystal floating-gate memory has the advantages of low power consumption and high erasing/programming speed [9,10,11]. The research on nanocrystal floating-gate memory is extensive, from traditional silicon germanium materials to third-generation semiconductor materials of SiC [15,16,17]. As reported by Jin et al. [15], floating-gate memory based on antimony-doped tin oxide nanoparticles has a maximum memory window of 85 V. However, there are 40 program/erase cycles, which has not been tried in 3D NAND devices. Lepadatu et al. carried out research on a new floating-gate MOS structure consisting of an HfO_2_/floating gate of a single layer of Ge QDs in the HfO_2_/tunnel HfO_2_/p-Si wafers [16]. The memory window of 3.8 V shows a very slow capacitance decease, which is not adopted in 3D NAND devices. According to the research of Andrzej et al. [17], a nanocrystalline SiC floating-gate memory exhibits better charge retention characteristics than the conventional floating-gate memory. However, the program/erase speed is in the scale of S, which is not applied in 3D NAND memory. Compared with the above nanocrystalline floating-gate memory, nc-Si-based floating-gate memory not only has faster program and erase speeds, but also has high durability for 10^7^ program/erase cycles. In particular, its high compatibility with modern microelectronics technology is beneficial to be integrated with computing in-memory chips. Our previous work focused on improving the density of 3D NAND flash memory by using double-layered nanocrystalline Si dots [18]. Up to now, there are few studies on how to improve the carrier injection efficiency of 3D NAND flash memory based on an nc-Si floating gate by novel design of the control gate thickness of the nano-silicon floating-gate memory.

In this paper, we first report that 3D NAND memories based on an nc-Si floating gate with high carrier injection efficiency could be obtained by novel design of the control barrier layer. The carrier injection efficiency can be monitored through the direction of capacitance–voltage (C–V) hysteresis of the nc-Si floating-gate MOS structure [19,20,21,22,23,24,25,26]. According to the principle of the floating-gate memory, the threshold voltage corresponding memory window can be expressed by the following equation.
(1)$$\Delta V_{th}=\frac{qn_{nc}}{\varepsilon_{ox}}\left(t_{cntl}+\frac{1}{2}\frac{\varepsilon_{ox}}{\varepsilon_{Si}}t_{nc}\right)$$

In the equation, $\Delta V_{th}$ is the threshold voltage shift, $t_{cntl}$ is the thickness of the control nitride, $t_{nc}$ is the size of a single nanocrystal, $\varepsilon_{ox}$ and $\varepsilon_{Si}$ are the dielectric constants of the SiO_2_ tunneling layer and nc-Si, respectively, *q* is the magnitude of electronic charge, and $n_{nc}$ is the density of the nanocrystals. According to the equation, the value of the memory window is related to the parameters of the floating gate, such as the density of the nc-Si, the thickness of the control layer, and the tunnel layer, etc. To improve the carrier injection efficiency, a novel design of the parameters is vitally important. Compared with the program/erase pulse cycle of an nc-Si floating-gate MOSFET, the C–V characteristic of an nc-Si floating-gate MOS structure can directly reflect the carrier injection efficiency without the influence of the source electrode and the drain electrode. Therefore, we focus on the C–V investigation of an nc-Si floating-gate MOS structure to improve the carrier injection in 3D nc-Si floating-gate memory. So far, the relationship between C–V hysteresis direction and carrier injection efficiency has been less reported in an nc-Si floating-gate memory [27,28]. Here, we first report that the C–V hysteresis direction of nc-Si floating-gate MOS structures could be shifted from counterclockwise to clockwise when the control layer thickness is reduced from 25 nm to 22 nm, which is influenced by the step size of the bias and the gate voltage [29]. When the thickness of the control layer reaches 25 nm, the direction of the C–V hysteresis can be maintained in the counterclockwise direction. However, when the thickness of the silicon nitride layer is reduced to 22 nm, a clockwise hysteresis curve will first appear in the low gate voltage scanning process. As the voltage increases, the window of the curve will gradually become smaller and then turn into a normal counterclockwise hysteresis curve. This is due to the carrier injection from the top electrode into the defect state of the SiN_x_ layer [30,31]. A thicker SiN_x_ control layer can block the transfer of carriers between the top electrode and the nc-Si dots, thereby improving the carrier injection efficiency from the Si substrate to the nc-Si dots. Combined with HRTEM and energy band theory, the relationship between the carrier injection efficiency and the C–V hysteresis direction is further revealed, which is beneficial to optimizing the performance of 3D NAND memory based on an nc-Si floating gate for future application in in-memory computing.

## 2. Materials and Methods

Figure 1 shows the schematic diagram of a 3D nc-Si floating-gate memory with three layers of a-Si:H channels. The substrate is a p-type silicon wafer with a crystal orientation of <100> and a resistivity of 6 to 9 Ω cm. To obtain the nc-Si floating-gate MOS structure, the Si substrate was first cleaned by standard RCA procedures [32]. The natural oxide layer grown on the surface was removed with dilute hydrofluoric acid. In order to fabricate the ultra-thin tunneling SiO_2_ layer, the thermal dry oxidation method under a temperature of 850 °C was employed [33,34]. The thickness of the tunnel SiO_2_ layer was 5 nm. Then, an a-Si layer was deposited on the tunnel SiO_2_ layer by introducing SiH_4_ into the PECVD chamber with an RF source frequency of 13.56 MHz [35,36]. The substrate temperature was 250 °C. Under the same circumstances, NH_3_ and SiH_4_ were decomposed to fabricate the SiN_x_ control layer on the surface of the a-Si:H layer. The thickness of the SiN_x_ control layer is 22 nm. To check the role of the SiN_x_ layer, a reference sample with an SiN_x_ thickness of 25 nm was fabricated by the same process. Finally, the samples were thermally post-treated at 1000 °C under ambient N_2_ to form nc-Si dots in the a-Si:H layer. For the convenience of the electric measurements, aluminum (Al) top electrodes were thermally evaporated on the surfaces of the samples with a shadow mask to form a circular spot. Al was thermally evaporated at the back side of the Si substrate as the bottom electrode. After the annealing process, the nc-Si embedded floating-gate MOS cross-sectional structure can be directly revealed by high-resolution cross-section transmission electron microscopy (HRTEM) with a JEM2010 electron microscope working at 200 kV. The C–V, transfer, and output characteristic were measured by using an Agilent B1500A at room temperature.

## 3. Results and Discussion

Figure 1a is a schematic diagram showing how the 3D nc-Si dots floating-gate flash memory based on three-layered a-Si:H channels is fabricated, according to the following steps. First, the surface of the Si substrate was covered by a thicker silicon oxide layer of 300 nm, which was grown by the wet oxidation method. Second, three layers of amorphous Si:H and SiO_2_ were alternately deposited on the surface of the SiO_2_ layer in the PECVD chamber at 300 °C. The thickness of the Si:H and SiO_2_ layers was 70 nm and 100 nm, respectively. Then, the three layers of a-Si:H and SiO_2_ were patterned by electron beam lithography and dry-etched to form the amorphous silicon channel. An nc-Si floating gate was grown on the surface of the three-layer a-Si channels, according to the preparation method of the MOS structure. In the following process, an a-Si:H terrace was obtained by etching to prepare for the drain electrode and the source electrode. The hole position of the drain electrode and the source electrode in each a-Si:H was different from each other, which was formed by etching of each a-Si:H terrace according to the pattern of the photolithography. Meanwhile, the holes of the source electrode and the drain electrode were obtained by etching the a-Si:H from the first to the third a-Si:H layers to expose the three a-Si:H layers. At last, the construction of the source electrodes and drain electrodes were completed by filling the holes of the source electrodes and drain electrodes with aluminum. Meanwhile, the metal of the gate electrodes was also deposited by thermal evaporation followed by lift-off technology to prepare for the electrical measurement. The C–V, erasing, and programming, as well as output characteristics, were measured by using an Agilent B1500A at room temperature. Under the bias of the gate, the nc-Si floating-gate unit on the two side walls of the 3D a-Si:H channel can be chosen to complete the writing and erasing functions separately. The site of the chosen nc-Si floating-gate cells decided by the position of the source electrode and the drain electrode, which range from the first layer to the third layer of the a-Si:H.

As revealed in Figure 1b, the nc-Si floating-gate MOS structure, including a tunnel oxide layer of 5 nm, an nc-Si layer of 3 nm, and a control layer of 22 nm, is clearly observed. It is evident that the nc-Si crystals of less than 3 nm are embedded in the a-Si sublayers, as presented in Figure 1c. A clear lattice image of nanocrystalline silicon can be seen, which corresponds to the crystal face index of (100). The C–V measurement diagram of the nc-Si floating-gate MOS device is shown in Figure 1d. During the C–V measurement, the top aluminum electrode is applied with the gate voltage, and the bottom aluminum electrode is grounded. During the double C–V measurement, the step size can be changed from 20 mV/s to 2500 mV/s. As illustrated in Figure 1e, the definition of the step size can be expressed by the Formula (2). It depends on the output voltage (V_step_) and the time of each scanning step, which can be divided into delay time (T_d_) and integration time (T_IN_).
(2)$$v=\frac{V_{\mathrm{step}}}{T_{d}+T_{\mathrm{IN}}}$$

We change the value of the T_IN_ to tune the step size. During the C–V measurement for the nc-Si floating-gate MOS structure device, we make the output voltage (V_step_) remain the same.

Figure 2a shows the double C–V characteristic curves of the SiN_x_/nc-Si/SiO_2_ floating-gate MOS structure under different step sizes, from 25 to 500 mv/s. The thickness of the control layer is 22 nm. The frequency is 1 MHz. The inset of each figure shows the variation of the flat band voltages (V_fb_s) with the bias increasing. As indicated in Figure 2a, under the step size of 20 mV/s, the direction of C–V hysteresis remains counterclockwise with the scanning bias increase from (−2 V, 2 V) to (−8 V, 8 V), and the maximum window is 1.14 V. It is interesting to find that a clockwise C–V hysteresis window appears under the bias from −2 V to 2 V when the step size increases to 250 mV/s, as shown in Figure 2b. When the bias changes from −3 V to 3 V, the clockwise window expands to a maximum of about 0.35 V. With the bias changing from −4 V to 4 V, the clockwise window begins to reduce. It is worth noting that the direction of the hysteresis curve changes to counterclockwise under the bias voltage from −5 V to 5 V. The counterclockwise window is continuously enhanced with the bias increasing from (−6 V, 6 V) to (−7 V, 7 V). The memory window reaches a saturation value of 0.93 V with the bias increasing from −8 to 8 V. After the step size is enhanced to 500 mV/s, the clockwise direction can be detected under the lower bias range from (−2 V, 2 V) to (−5 V, 5 V). In contrast with the step size of 250 mV/s, the critical bias corresponding to the direction transition from clockwise to counterclockwise is increased from −6.3 V to 6.3 V, as shown in Figure 2c.

Figure 3 shows the schematic diagram of the counterclockwise C–V hysteresis formation from the SiN_x_/nc-Si(a-Si)/SiO_2_ floating-gate MOS structure. As displayed in Figure 3a,b, the free electrons accumulated on the surface of the Si substrate under a positive bias. With the positive bias reaching a critical value, the electrons can tunnel through the SiO_2_ layer into the nc-Si layer, as shown in Figure 3c. The transfer of electrons contributes to a positive movement of the flat band voltage. The control SiN_x_ layer can prevent electrons moving from the nc-Si to the upper electrode just like a blockade. As a result, the electrons can be stored in the nc-Si quantum dots in the nc-Si layer. Similarly, when the gate voltage changes from positive to negative, the free holes tunnel from the substrate into the nc-Si layer, causing the flat band voltage to move negatively, as shown in Figure 3d–f. Therefore, a counterclockwise C–V hysteresis window is formed.

In an attempt to provide insight into the formation mechanism of the clockwise C–V hysteresis, the C–V characteristics of the SiN_x_/nc-Si(a-Si)/SiO_2_ floating-gate MOS structure with a thicker SiN_x_ control layer of 25 nm were tested, as shown in Figure 4. It is interesting to find that the clockwise C–V window cannot be observed with the step size increasing from 250 mV/s to 2500 mV/s, which proves that the clockwise C–V hysteresis is related to the thickness of the SiN_x_ control layer. When the SiN_x_ control layer is thinner, the corresponding barrier is lower, which is easier for holes moving from the top electrode to the defects in the nitride SiN_x_ control. In particular, the defects in the SiN_x_ control have lower energy levels than the Fermi level of the aluminum, and the number of the holes injected into the defects is higher than that of the electrons tunneling from the Si substrate to the nc-Si layer under the same gate bias. Thus, the holes from the top electrode become the main contributor to the charge movement, which determines that the movement direction of the flat band voltage is reverse that induced by the electrons from the substrate. Therefore, we can detect the formation of the clockwise C–V hysteresis.

To further reveal the relationship between the carrier injection efficiency and the direction of the C–V memory window, we elaborate the energy band gap model of the SiN_x_/nc-Si/SiO_2_ floating-gate MOS structure with thicker and thinner SiN_x_ control layers under different step sizes of bias [28,37]. As presented in Figure 5a–c, when the SiN_x_ control layer of the nc-Si floating-gate MOS structure is thicker, a larger quantity of electrons tunnel from the Si substrate to the nc-Si layer under the positive bias. The higher barrier of the thicker SiN_x_ control layer can reduce the movement of holes from the top electrode to the SiN_x_ control layer. Therefore, the number of the electrons tunneling from the substrate to the nc-Si layer is larger than that of the holes stored in the defects of the SiN_x_ control layer from the Al top electrode. The electrons tunnelling from the substrate to the nc-Si layer results in the positive movement of the flat band voltage. Under the negative bias, the number of holes tunneling from the substrate to the nc-Si layer is larger than that of the electrons stored in the defects of the SiN_x_ control layer from the Al top electrode. The holes tunnelling from the substrate to the nc-Si layer result in the negative movement of the flat band voltage. Therefore, a counterclockwise hysteresis is formed. Even with the step size of the bias increasing from 250 to 2500 mV/s, the counterclockwise direction of the C–V memory window remains unchanged, as revealed in Figure 4a. When the SiN_x_ control layer is thinner, the lower barrier height of the SiN_x_ control layer is easier for the carriers moving from the top electrode to the SiN_x_ control layer, as displayed in Fig5.d-f. When the step size is 50 and 250 mV/s, the bias duration on the electrode is shorter. Under the lower bias, the number of the carriers tunneling from the Si substrate to the nc-Si layer is smaller than that of the carriers moving from the top electrode to the SiN_x_ control layer, which results in the reverse movement of the flat band. Thus, the clockwise memory window can be observed. It is worth noting the number of carriers tunnelling from the substrate to the nc-Si can be enhanced with the bias increasing. Under the lower bias, it is larger than the number of the carriers moving from the top electrode to the SiN_x_ control layer, so the direction of the C–V memory window will change to counterclockwise, as shown in Figure 5i,j. This is the reason why the clockwise memory window shows a changeable trend from the maximum to the minimum, as proved by Figure 4b,c. When the step size is reduced to 20 mV/s, the bias duration on the electrode gets longer. The number of the carriers tunneling from the substrate to the nc-Si layer is larger than that of the carriers stored in the defects of the SiN_x_ control layer from the Al top electrode. The counterclockwise memory window can be maintained from the low bias to the high bias, as shown by Figure 4a. The contribution of the SiN_x_ layer to the C–V hysteresis window can also be illustrated by comparing the two devices with different thickness of the SiN_x_ layer, which is shown in Figure 3b and Figure 4a. Under the same step size of 250 mV/s, the memory window of the device with a thickness of 25 nm is 2.5 V, which is larger than the 1 V of the device with a thickness of 25 nm. Because the barrier height of the SiN_x_ layer can be enhanced with the thickness increasing, it is more difficult for carriers to move from the top electrode to the defect in the SiN_x_. As a result, the carrier injection efficiency from the Si substrate into the nc-Si dots is enhanced, which leads to a larger memory window. Based on the above analysis, the thicker SiN_x_ control layer is beneficial to increasing the carrier injection efficiency.

The erasing and programming performance of the 3D NAND nc-Si floating-gate memory unit, based on a three layered a-Si:H channel with a thicker control SiN_x_ of 25 nm, is shown in Figure 6a,b. As displayed in Figure 6c, the corresponding output current is increased with the positive gate voltage changing from 1 to 5 V, displaying the N-type characteristic of the a-Si channel. A stable memory window of 1.21 V can be obtained under the programming and erasing voltages of +7 V and −7 V. The programming and erasing speed reaches 100 µs. It is worth noting that the stable memory window of 1.21 V can be maintained after 10^7^ P/E cycles, as shown in Figure 6d. The novel performance of the 3D NAND nc-Si floating-gate memory unit is determined by the design of the nc-Si floating gate with a novel SiN_x_ control layer.

## 4. Conclusions

In summary, carrier injection efficiency can be controlled in 3D NAND memory based on a nanocrystalline silicon floating gate by novel design of the control barrier layer. The C–V hysteresis direction of the nc-Si floating-gate MOS structure can change from counterclockwise to clockwise when the thickness of the control layer decreases from 25 to 22 nm, which is easily effected by the step size of the bias and gate voltage. As the thickness of the control layer reaches 25 nm, the direction of the C–V hysteresis remains counterclockwise. The thicker SiN_x_ control layer can block the transfer of carriers between the top electrode and the nc-Si dots, leading to enhanced carrier injection efficiency from the Si substrate to the nc-Si dots. HRTEM and the energy band theory model further reveal the relationship between the carrier injection efficiency and the direction of the C-V hysteresis, which is beneficial to optimizing the performance of 3D NAND memory based on an nc-Si floating-gate memory for the application of in-memory computation.

## Figures and Tables

**Figure 1 nanomaterials-13-00962-f001:**
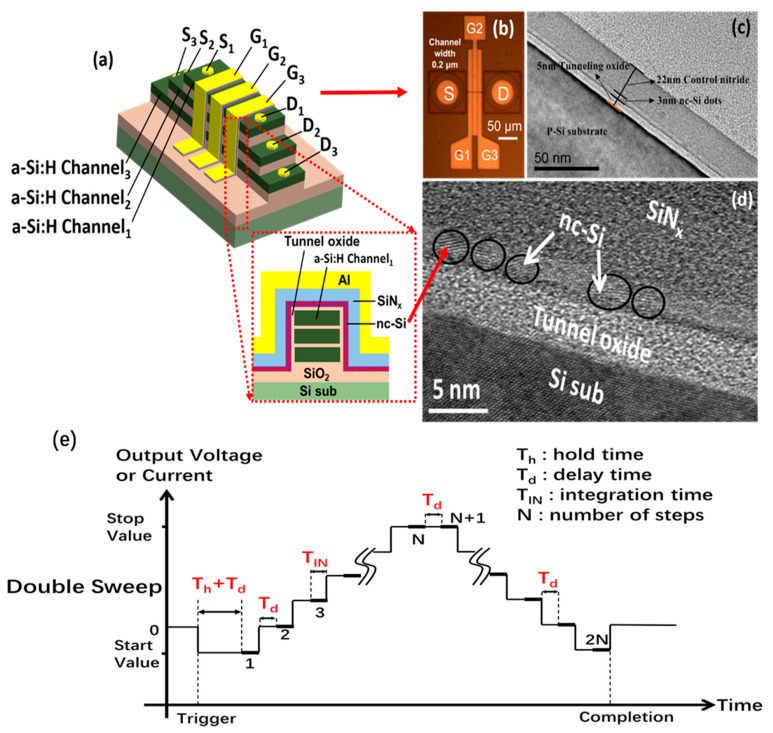
(**a**) Schematic diagram of 3D flash memory based on three-layered a-Si:H channels with a single-layered nc-Si dots floating-gate structure. (**b**) The optical image of 3D NAND flash memory based on the SiN_x_/nc-Si(a-Si)/SiO_2_ floating gate. (**c**,**d**) Ordinary and high resolution cross-section TEM photograph of the SiN_x_/nc-Si(a-Si)/SiO_2_ floating-gate MOS structure. (**e**) The time dependence of output voltage for double C–V measurement.

**Figure 2 nanomaterials-13-00962-f002:**
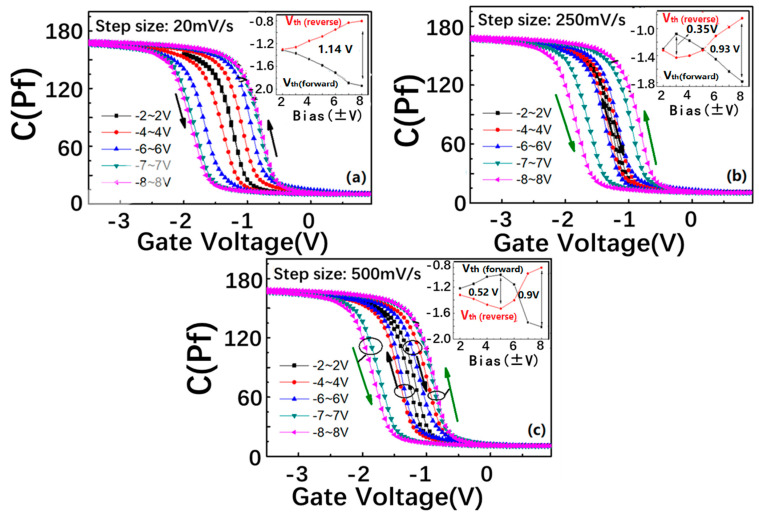
Double C–V sweeps of the SiN_x_/nc-Si(a-Si)/SiO_2_ floating-gate MOS structure in different scanning bias ranges. The step sizes are (**a**) 20 mV/s, (**b**) 250 mV/s, and (**c**) 500 mV/s, respectively. The insets show the scanning bias-dependent V_fb_ under different step sizes.

**Figure 3 nanomaterials-13-00962-f003:**
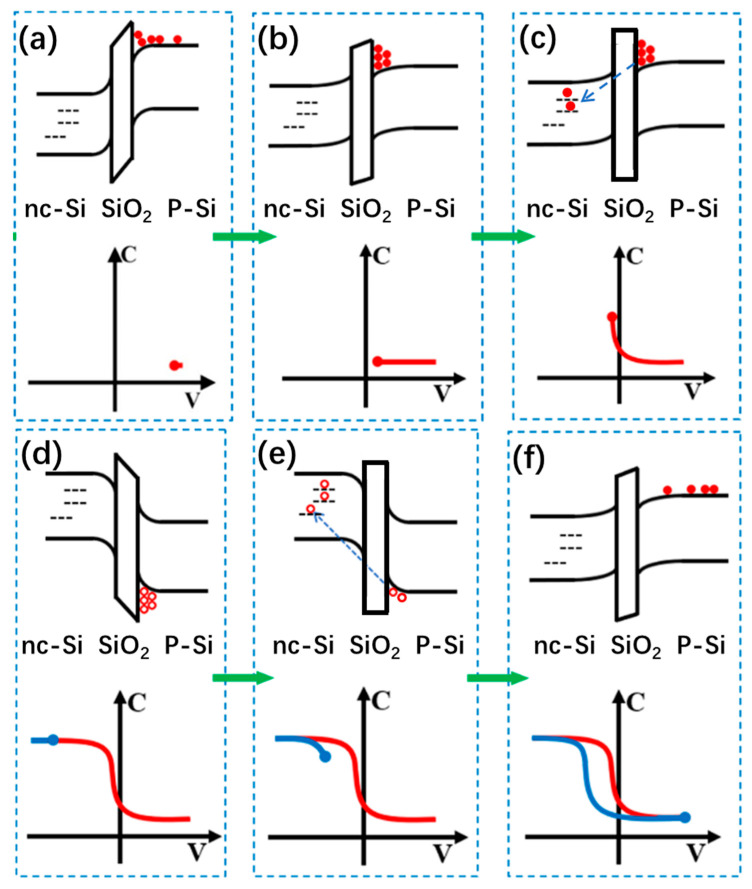
Formation mechanism of the counterclockwise C–V hysteresis of the SiN_x_/nc-Si(a-Si)/SiO_2_ floating-gate MOS structure. (**a**–**c**) Under the positive gate voltage, the electrons tunnel from the substrate into the nc-Si layer, leading to the positive movement of the flat band voltage as shown by the red lines. (**d**–**f**) Under the negative gate voltage, the holes tunnel from the substrate into the nc-Si layer, leading to the negative movement of the flat band voltage as shown by the blue lines.And a counterclockwise C–V window is formed. The hollow circles and the solid circles represent the holes and electrons, respectively. The blue arrow represents the direction of electron tunneling and hole tunneling.

**Figure 4 nanomaterials-13-00962-f004:**
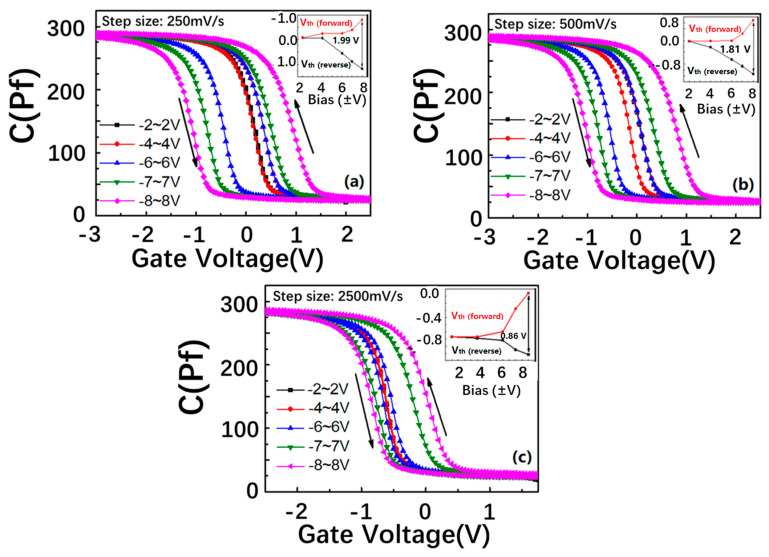
Double C–V sweeps of the fabricated SiN_x_/nc-Si(a-Si)/SiO_2_ floating-gate MOS structure with a thicker nitride SiN_x_ control layer in different scanning bias ranges, and the ramp rates are (**a**) 250 mV/s, (**b**) 500 mV/s, and (**c**) 2500 mV/s, respectively. The insets show the scanning bias range is dependent on the V_fb_s at different ramp rates.

**Figure 5 nanomaterials-13-00962-f005:**
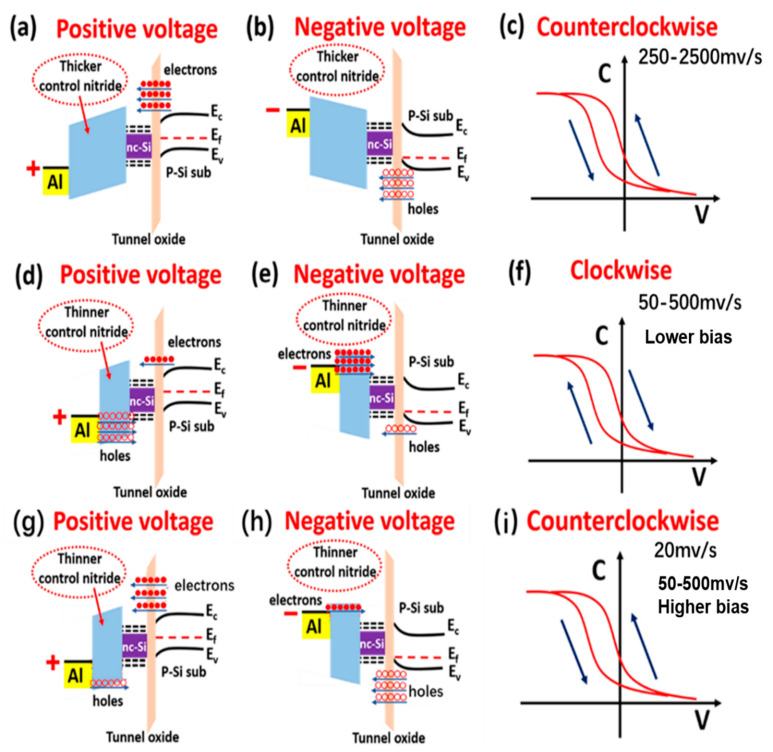
The energy band gap model of the SiN_x_/nc-Si/SiO_2_ floating-gate MOS structure with thicker SiN_x_ control layer under (**a**) positive and (**b**) negative voltages. (**c**) The counterclockwise hysteresis memory window of the SiN_x_/nc-Si/SiO_2_ floating-gate MOS structure with step size from 250 to 2500 mv/s. The energy band gap model of the SiN_x_/nc-Si(a-Si)/SiO_2_ floating-gate MOS structure with the thinner control SiN_x_ layer under (**d**) positive and (**e**) negative voltages. (**f**) The clockwise C–V hysteresis memory window of the SiN_x_/nc-Si/SiO_2_ floating-gate MOS structure with the step size from 50 to 500 mv/s. The energy band gap model of the SiN_x_/nc-Si(a-Si)/SiO_2_ floating-gate MOS structure with the thinner SiN_x_ control layer under (**g**) positive and (**h**) negative voltages. (**i**) The counterclockwise C–V hysteresis memory window of the SiN_x_/nc-Si/SiO_2_ floating-gate MOS structure with the step size of 20 mv/s. The hollow circles and solid circles represent the holes and electrons, respectively.

**Figure 6 nanomaterials-13-00962-f006:**
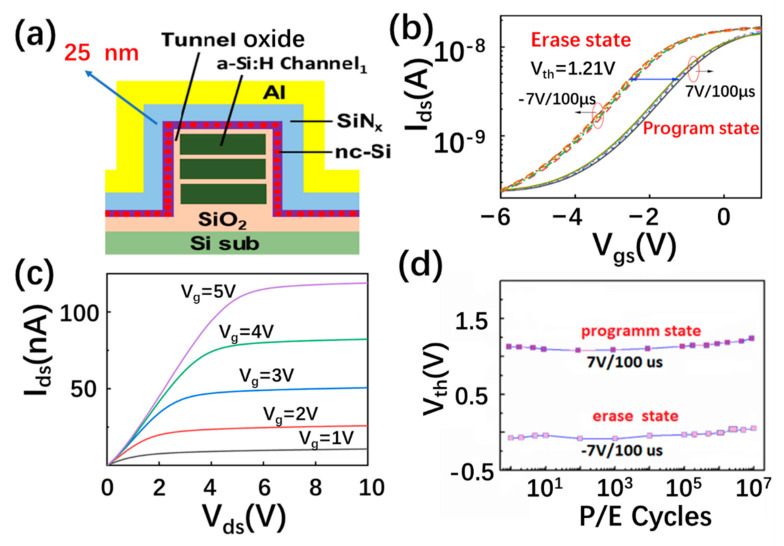
(**a**) The schematic diagram of the nc-Si floating-gate memory with a SiN_x_ thickness of 25 nm. (**b**) The erasing and programming characteristics of the 3D nc-Si floating-gate memory with a SiN_x_ thickness of 25 nm. The programming and erasing speeds are +7 V/100 us and −7 V/−100 us (**c**) The output characteristics of the 3D nc-Si floating-gate memory with a SiN_x_ thickness of 25 nm. (**d**) The endurance characteristics of the 3D nc-Si floating-gate memory with a SiN_x_ thickness of 25 nm after the 10^7^ P/E cycle operations at +7 V/100 us and −7 V/100 µs.

## Data Availability

Data can be available upon request from the authors.

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
