# Peer review of "Controlling the Carrier Injection Efficiency in 3D Nanocrystalline Silicon Floating Gate Memory by Novel Design of Control Layer"

_nanomaterials, 2023, doi:10.3390/nano13060962_

Round 1

Reviewer 1 Report

1) what does it mean the step size change in Fig. 2? I couldn't see the number of points change between Fig. 2a, b, and c. And it is curious that fast-sweeping (like 2500mV/s) can capture all the C-V data points correctly without any noise / overshoot.

2) Are S1, S2, and S3 in contact in Fig. 1? (Same question for D1, D2, and D3)

3) can you explain how the electrons tunnel through the oxide in Fig. 3c? since there's no electric field at oxide layer, it looks like that electrons don't have energy to tunnel through even though electrons are piled up at the oxide interface.

4) Please address how the nanocrystal can be used in 3D NAND flash structure, not this 3-layer stack in the discussion and conclusion section.

5) Why is this experimented in C-V, not program/erase pulse cycle?

Reviewer 2 Report

The authors address a topical research issue related to obtaining a 3D NAND flash memory based on three-layered a-Si:H channels with a single-layered nc-Si dots floating gate. They explain how C–V hysteresis direction of nc-Si floating gate MOS structure can transfer from counterclockwise to clockwise when the thickness of the control layer decreases from 25 to 22 nm, which is easier affected by the step size of bias and gate voltage. The manuscript demonstrates good innovation.

It is very difficult to determine the exact transport mechanism in such a structure, such as the level of insulation provided by the SiN layer and the types of defects present. However, the authors have provided explanations that are highly probable.

Therefore, I recommend that this paper can be accepted for publication after appropriate revisions:

- the manuscript is a development of the article recently published by the group (Xinyue Yu, et al. Nanomaterials 2022, 12, 2459) and at least in the introduction part it shows a small degree of similarity. As a suggestion/comment, the authors could write the introduction without following the structure of the previously published article.

- after the removal of the native oxide, the Si surface becomes very reactive. How do you protect this surface between the two technological processes (native SiO2 cleaning and controlled SiO2 layer growing?

- is Figure 1(b) a TEM image?

Reviewer 3 Report

Authors studied the nanoscrystalline silicon with direct application in 3D NAND memory. Present study may be of interest but an extensive revision is required.

My concerns are as follows:

1) authors should clearly state novelty of their work in comparison with their previous study (https://doi.org/10.3390/nano12142459). Present work and previous work are having a lot of similarities (e.g., figures 1 and 2)

2) Introduction should be more in deep. Detailed description in the field is not given. 

3) Figure 3 quality must be improved.

4) Discussion should be more in deep.

5) English of the manuscript requires to be enhanced. Many sentences are hard to read and understand.

For example, abstract line 17-18 sentence "Here we first..." contains two active verbs and is hard to understand.

line 44 should be "has always been concern for"

page 2

line 65  "e`s" should be  "e stands for the permeability" - then authors should also explain ox and Si subsripts. 

page 3

thermal dry oxidation method - please add reference. same for other methods used to prepare sample.

page 9

line 225 - "Therefore, the ..." should be "hence" or "thus" instead.

etc.

Overall an extensive work is needed to justify publication of this work.

Round 2

Reviewer 1 Report

No comments 

Reviewer 3 Report

Authors have revised manuscript as requested. As a result, I can recommend it to be publised.